



# 1 Influence of low-frequency variability on high and low groundwater

# 2 levels: example of aquifers in the Paris Basin

Lisa Baulon[1,2], Nicolas Massei[1], Delphine Allier[2], Matthieu Fournier[1], Hélène Bessiere[2]
[1]Normandie Univ, UNIROUEN, UNICAEAN, CNRS, M2C, 76000 Rouen, France
[2]BRGM, 3 av. C. Guillemin, 45060 Orleans Cedex 02, France
*Correspondence to*: Lisa Baulon (lisa.baulon@etu.univ-rouen.fr)
**Abstract.** Groundwater levels (GWL) very often fluctuate over a wide range of timescales (infra-annual, annual, multi-
annual, decadal). In many instances, aquifers act as low-pass filters, dampening the high-frequency variability and
amplifying low-frequency variations (from multi-annual to decadal timescales) which basically originate from large-scale
climate variability. In the aim of better understanding and ultimately anticipating groundwater droughts and floods, it
appears crucial to evaluate whether (and how much) the very high or very low GWLs are sensitive to such low-frequency
variability (LFV), which was the main objective of the study presented here. As an example, we focused on exceedance and
non-exceedance of the 80% and 20% GWL percentiles respectively, in the Paris Basin aquifers over the 1976-2019 period.
GWL time series were extracted from a database consisting of relatively undisturbed GWL time series regarding
anthropogenic influence (water abstraction by either continuous or periodic pumping) over Metropolitan France. Based on
this dataset, our approach consisted of exploring the effect of GWL low-frequency components on threshold exceedance and
non-exceedance by successively filtering out low-frequency components of GWL signals using maximum overlap discrete
wavelet transform (MODWT). Multi-annual (~7-yr) and decadal (~17-yr) variabilities were found to be the predominant
LFVs in GWL signals, in accordance with previous studies in the northern France area. Filtering out these components
(either independently or jointly) to (i) examine the proportion of high level (HL) and low level (LL) occurrences generated
by these variabilities, (ii) estimate the contribution of each of these variabilities in explaining the occurrence of major
historical events associated to well-recognized societal impacts.
A typology of GWL variations in Paris Basin aquifers was first determined by quantifying the variance distribution across
timescales. Four GWL variation types could be found according to the predominance of annual, multi-annual or/and decadal
variabilities in these signals: decadal dominant (type iD), multi-annual and decadal dominant (type iMD), annual dominant
(type cA), annual and multi-annual dominant (type cAM). We observed a clear dependence of high and low GWL to LFV
for aquifers exhibiting these four GWL variation types. In addition, the respective contribution of multi-annual and decadal
variabilities in the threshold exceedance varied according to the event. In numerous aquifers, it also appeared that the
sensitivity to LFV was higher for LL than HL. A similar analysis was conducted on the only available long-term GWL time
series which covered a hundred years. This allowed us to highlight a potential influence of multidecadal variability on HL
and LL too.
This study underlined the key role of LFV in the occurrence of HL and LL. Since LFV originates from large-scale stochastic
climate variability as demonstrated in many previous studies in the Paris Basin or nearby regions, our results point out that i)
poor representation of LFV in General Circulation Models (GCM) outputs used afterwards for developing hydrological
projections can result in strong uncertainty in the assessment of future groundwater extremes (GWE), ii) potential changes in
the amplitude of LFV, be they natural or induced by global climate change, may lead to substantial changes in the
occurrence and severity of GWE for the next decades. Finally, this study also stresses the fact that due to the stochastic
nature of LFV, no deterministic prediction of future GWE for the mid- or long term horizons can be achieved even though
LFV may look periodic.
**1.  Introduction**
The knowledge of hydroclimatic extremes is a major concern in a global change context. Hydroclimatic extremes, leading to
droughts or floods, can have major consequences on our societies. During hydrological drought periods, many restrictions of
water uses can be imposed to the population. For instance each summer in France, regular restrictions are imposed due to
hydrogeological droughts (Maréchal and Rouillard, 2020). These restrictions are damaging especially for agricultural and



industrial activities. Floods are equally harmful, the best-known example across France is the flooding of the Somme River
Basin in 2001. This flooding cost 100 million euros of damages and 1100 people were evacuated due to the event long
duration spreading over several months (Deneux and Martin, 2001). Moreover, flooding events can also lead to other
damages such as erosive events or the degradation of water quality.
The hydrological droughts are characterised by below-normal groundwater levels (GWL) or water levels in lakes, declining
wetland area and decreased streamflow (Van Loon, 2015). Surface and subsurface water resources are then inadequate for
established water uses of a given water resources management system (Mishra and Singh, 2010). Many studies about
hydrological droughts focus on streamflow, but it is equally important to look at aquifers and GWL. Consequently, Mishra
and Singh (2010) introduced groundwater drought as a new type of drought alongside the four main types of droughts:
meteorological drought, hydrological drought, agricultural drought, socioeconomic drought. Groundwater droughts occur on
time scales from months to years (Van Lanen and Peters, 2000). They follow periods of precipitation deficits combined with
high evapotranspiration rates, that in turn causes low soil moisture content and low groundwater recharge. Abstraction of
groundwater can also enhance naturally occurring droughts (Van Lanen and Peters, 2000). Groundwater droughts may also
have consequences on streamflow with less support of the water table during low river flow periods. This can be highly
problematic especially in catchments where rivers are strongly sustained by water tables, such as the Seine river where 81%
of the flow is supported by groundwater (Flipo et al., 2020).

Contrastingly to hydrological droughts, that are processes setting up slowly, floods are fast phenomena resulting from
extreme precipitation, snowmelt and high initial soil moisture (Berghuijs et al., 2016; Wasko and Nathan, 2019; Bertola et
al., 2021). In case of GWL, the speed of emergence of extremely high levels can vary greatly according aquifers and the
GWL variation type. In reactive systems, the water table quickly reacts to an exceptional rainy winter leading to high GWL
at the end of the recharge period. In inertial systems, several years of exceptional rainfall and recharge are needed to reach
particularly high levels. In both cases, extremely high GWL can lead to groundwater flooding. A key example in France was
the 2001 floodings in the Somme region that were the consequence of exceptional GWL (higher than the soil surface) and



exceptional levels of Somme river. These floodings were the result of above-average winter rainfall during several years
rising GWL in the chalk – with limited summer recession of GWL – and an exceptional previous winter with strong
precipitation leading to rapidly increase levels of 10m generating the disastrous floodings (Deneux and Martin, 2001; Habets
et al., 2010). This 2001 event arised from both a low-frequency variability (LFV) with a slow increase of GWL during
several years and a high-frequency variability that is superimposed, with a sudden rise of GWL during the 2000-2001 winter
(Pointet et al., 2003).

It is expected with the climate change to observe increasingly hydroclimatic extremes with stronger intensities and/or higher
frequencies (IPCC, 2012; Hirabayashi et al., 2013; Tramblay et al., 2020). Therefore in the litterature, studies on
hydroclimatic extremes deal very often with trend analyses to identify potential increases in the frequency of extremes and
their intensity (Hodgkins et al., 2017; Mangini et al., 2018; Blösch et al., 2019; Vicente-Serrano et al., 2021). To describe the
long-term evolution of hydroclimatic extremes and their characteristics (e.g., duration, magnitude, intensity), meteorological
or hydrological drought indices such as the Standardised Precipitation Index (SPI; McKee et al., 1993), Standardised
Precipitation Evapotranspiration Index (SPEI; Vicente-Serrano et al., 2010), Standardised Streamflow Index (SSI; Vicente-
Serrano et al., 2012), Standardised Groundwater level Index (SGI; Bloomfield and Marchant, 2013), are commonly used.
These indices are widely used to detect trends in meteorological or hydrological droughts and their variability across time
(Vicente-Serrano et al., 2021). However, although these indices are useful tools to describe droughts, their principal limit
arises from the standardisation allowing for spatial comparison but therefore hindering to keep the variance notion in time
series. For instance, this can be particularly limiting to understand the emergence of high and low GWL whose amplitude
seems highly dependent of the maximum water level fluctuation.

In addition, the long-term effects of climate change on meteorological and hydrological variables may be modified by the
internal climate variability leading to their amplification, attenuation, or inversion (Fatichi et al., 2014; Gu et al., 2019).
Therefore, it is crucial to better understand the large-scale origin of these LFV variabilities and how catchments can filter
and modify them, in particular for prediction purposes. In this regard, Gudmundsson et al. (2011) indicated that the LFV of





runoff directly originates from the large-scale atmospheric circulation, while the catchments properties control the proportion
of variance of LFV in hydrological variables. Simultaneously, a large amount of studies adressed the large-scale origins of
such variabilities in hydroclimatic variables (streamflow, precipitation, groundwater, temperature), using climate indices and
atmospheric fields (Massei et al., 2010; Boé and Habets, 2014; Dieppois et al., 2013; Dieppois et al., 2016; Massei et al.,
2017; Neves et al., 2019; Liesch and Wunsch, 2019).

In northern France, more particularly in the Seine watershed, many studies highlighted ~7-yr and ~17-yr variabilities in
precipitation and streamflow (Massei et al., 2007; Massei et al., 2010; Fritier et al., 2012; Massei and Fournier, 2012;
Dieppois et al., 2013; Massei et al., 2017). Since then, CaWaQS model calibration – that is a Seine Basin Model – is
achieved on 17-year period as it is proved that groundwater and river water storage are stationary over such period (Flipo et
al., 2012; Flipo et al., 2020). The North Atlantic Oscillation (NAO) was described as one significant driver of such temporal
signature (~7-yr and ~17-yr) in precipitation and streamflow (Massei et al., 2007; Massei et al., 2010). Later, Massei et al.
(2017) highlighted using a composite analysis with Sea Level Pressure (SLP) that the atmospheric pattern associated to the
~7-yr variability was not exactly reminiscent of the NAO, with centers of action actually shifted to the North. Similarly, the
pattern associated to ~17-yr variability (called "~19.3-yr component" in Massei et al. (2017) study) was a spatially extended
pattern across the Atlantic ocean with lower SLP roughly following the Gulf Stream front. This result highlighted that
atmospheric patterns associated to ~7-yr and ~17-yr variabilities are not similar and these atmospheric patterns exhibit
centers of action that are not necessarily corresponding to those of established climate indices such as the NAO.

Aquifers very often act as low-pass filters, leading to high-amplitude LFV in GWL. Numerous studies also adressed the
physical and morphometric parameters controlling the significance of these variabilities in GWL: the superficial formations
properties, the vadose zone properties and the aquifers intrinsic properties being the main accountable of their magnitude
(Slimani et al., 2009; El Janyani et al., 2012; Velasco et al., 2017; Rust et al., 2018). In Normandy, Slimani et al. (2009) and
El Janyani et al. (2012) identified a significant ~7-yr variability in GWL of chalk aquifer. Recently, Baulon et al. (2020) also
identified a ~17-yr variability in significant proportion in GWL of some French northern aquifers constituted of chalk and





limestones. Therefore, aquifers exhibiting a significant LFV would display highly dependent extreme levels to such
variabilities. For instance, Rust et al. (2019) showed that hydrogeological droughts in UK are highly dependent of the ~7-yr
variability: the major droughts emerging during low multi-annual levels, excepting the 1975 drought. In addition, Bonnet et
al. (2020) described the influence of multi-decadal variability on high and low flows and how it can impact short-term
drought events through groundwater-river exchanges in the Seine basin.

In summary, a few studies concluded to a potential high impact of large-scale climate-induced LFV in supporting high and
low GWL, but none of them have yet investigated how much groundwater extremes (GWE) depend on such variability.
Throughout the text, for the sake of clarity, GWE will refer to both very high or very low GWL, according to certain
thresholds that will be defined in subsequent sections.

To answer this question, a simple approach based on the decomposition of GWL time series of northern French aquifers into
high- to low-frequency components is proposed in this study over the 1976-2019 period. Beforehand, we quantified the
variance distribution across timescales for assessing the significance of low-frequency variations in GWL signals,
particularly of multi-annual (~7-yr) and decadal (~17-yr) variabilities (Section 4). Then, our methodology consists to
evaluate the influence of timescales corresponding to multi-annual and decadal variabilities by filtering one or both
timescale(s) from the original signal to assess how their withdrawal affects threshold exceedance. First, we propose
estimating the proportion of high (HL) or low levels (LL) associated to the multi-annual or decadal variabilities, and then by
the combination of both (Section 5). We also propose determining through a long groundwater time series (106 years of data
available) if proportions of HL and LL associated to LFVs are consistent with those obtained in the short term. Second, we
propose determining on four well-known historical events the contribution of multi-annual and decadal variabilities in the
amplitude of threshold exceedance (ATE) and identify what parameters may control this contribution (Section 6).



## 2. Effective precipitation and groundwater data

For this study, we used 78 boreholes in the Paris Basin (northern France), with GWL time series being little or not affected by pumping (Fig. 1). The study area was restricted to the Paris Basin in order to carry out the analysis over a relatively long period (1976-2019). Boreholes were selected from a BRGM database on boreholes not influenced by human activities (Baulon et al., 2020) that was constituted in three steps:

(i)    a selection of boreholes with time series satisfying criteria of duration, minimum amount of data per month, maximum length of gaps;

(ii)    the crossing of pre-selected boreholes with other BRGM databases on known anthropogenic influences;

(iii)    numerous visualisations of time series with the hydrogeologists responsible for piezometric networks, in order to retain only non-influenced boreholes.

Time series of boreholes in this database were initially gathered in the ADES database that contains all groundwater data (quantity and quality) across continental France (https://ades.eaufrance.fr/).

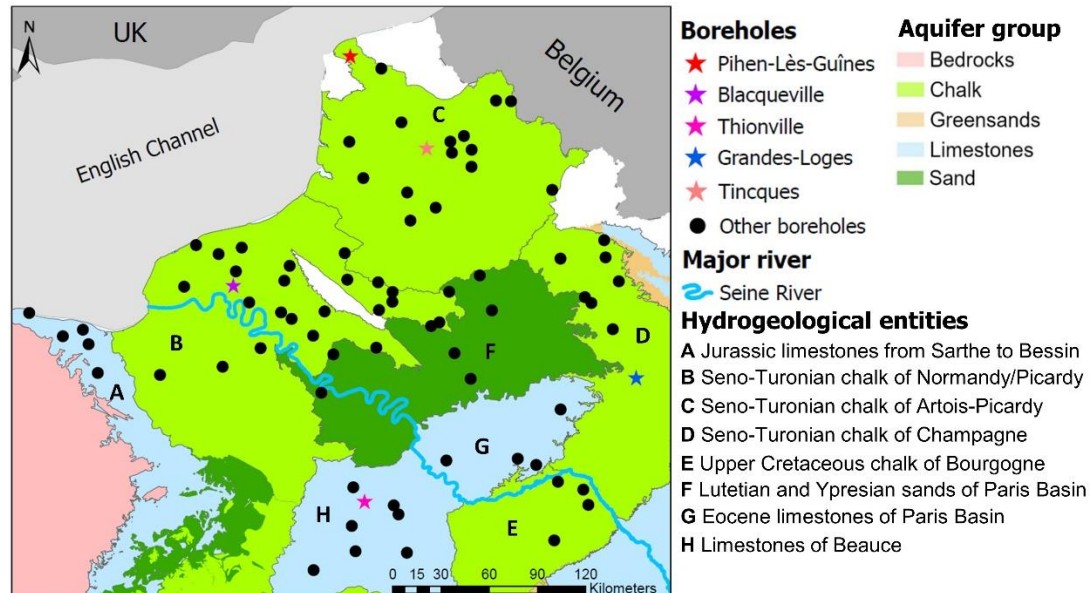

**Figure 1: Spatial distribution of the 78 selected boreholes through major hydrogeological entities of the Paris Basin.**



The criteria satisfied for selecting GWL time series for the present study were:

•        The length of time series must be higher or at least equal to 44 years.

•        The minimum amount of data within a month is set to one monthly datum before the measurement

automation and three data after the measurement automation.

•        The length of consecutive gaps must be <3 yr for time series starting after 1950 and <10 yr for time series

starting before 1950. This allows time series in the new database to preserve LFV in the data. Several gaps

in the time series can be allowed if these criteria are respected, and if the number of gaps and their lengths

are small.

Before data analysis, a visual check of the GWL time series served to remove or correct erroneous data.

The analysis of the influence of LFV on HL and LL was conducted over the 1976-2019 period providing the best
compromise between the spatial distribution of boreholes and time series length. In this work, all time series had a monthly
time step to consider all possible GWL variation types from most reactive to most inertial ones. Then, monthly missing
values were filled with linear interpolation to perform spectral analyses.

GWL time series capture chalky formations, calcareous formations and sandy formations of the Paris Basin. In addition, we
also used the GWL time series of Tincques monitoring the Seno-Turonian chalk of Artois-Picardy since 1903 (Fig. 1). This
time series allowed us to perform our analysis on a longer temporal perspective and compare results with those obtained on
short-term time series.

Time series of 4 boreholes, among the 78 of the study, were used to introduce the different GWL variation types in Paris
Basin aquifers: Pihen-Lès-Guînes, Blacqueville, Thionville, Grandes-Loges (Fig. 1). These boreholes were chosen because
each of them is representative of a GWL variation type.





For the aforementioned boreholes monitoring GWL of chalk aquifers (Pihen-Lès-Guînes, Blacqueville and Grandes-Loges),
we also investigated effective precipitation (EP) corresponding to these boreholes. For each borehole, we assigned to it the
mesh with EP time series the most explanatory of GWL. This linkage between the borehole and a SAFRAN mesh was
carried out prior to this study by correlative analysis. In this study, we used monthly cumulative EP time series over the
1976-2019 period.

The meteorological data (precipitation (P), snow, temperature and Penman-Monteith potential evapotranspiration (PET))
from the SAFRAN reanalysis were used as input data to compute effective precipitation. This reanalysis provides daily data
on a 8x8km2 mesh covering France from 1958 to 2019 (Vidal et al., 2010). The effective precipitation (EP = P – PET) were
computed using a gridded water budget model with 8km resolution at daily time step. It relies on the water budget method
proposed by Edijatno and Michel (1989). The water-budget method considers that in the water cycle, the soil acts as a
reservoir characterized by its water storage capacity. Edijatno and Michel (1989) introduced a quadratic law to progressively
empty the soil water reserves and to distribute the positive difference between P and PET between EP and soil storage.
**3.  Methodological approach**
**3.1. Characterization of groundwater multi-timescale variability**
In order to determine the prominence of LFV in GWL, the maximum overlap discrete wavelet transform (MODWT) analysis
was applied. This is an iterative filtering of the time series that uses a series of low-pass and high-pass filters. Consequently,
one high-frequency component called "wavelet detail" and one lower frequency component called "approximation" or
"smooth" are produced at each timescale. At the next level, the smooth is then subsequently decomposed into another
wavelet detail and smooth. The original signal can be rebuilt by summing up all wavelet details and the last smooth. The
original signal is then separated into a relatively small number of wavelet components from high- to low frequencies, which
together explain the total variability of the signal. Here, we achieved a full decomposition of the time series by applying the
filter bank up to a level corresponding to the log2($N$) where $N$ is the length of the time series. The least-asymmetric





(symmlet) wavelet "s20" was used in order to better capture variability at all timescales of sometimes relatively smooth
groundwater level time series.

However, unlike DWT, MODWT was essentially designed to prevent phase shifts in the transform coefficients at all scales
by avoiding downsampling – reducing by a factor 2 the number of coefficients – the signal with increasing scales. It results
that the computed wavelet and scaling coefficients at each scale remain aligned with the original time series; that is, the
variance explained by these coefficients is located where it truly lies in the time series analysed (Percival and Walden, 2000;
Cornish et al., 2003; Cornish et al., 2006). While not necessarily essential for signal or image processing or numerical
compression, this property is fundamental for physical interpretation of the wavelet details in multiresolution analysis, and
has already been used to that purpose in several studies such as Percival and Mofjeld (1997), Massei et al. (2017) and Pérez
Ciria et al. (2019).

The dominant frequency associated with each MODWT wavelet detail was calculated by Fourier transform of each wavelet
detail. The MODWT also provides the amount of variance (or energy) explained by each wavelet detail and frequency level.
The energy percentage of a given wavelet detail expresses the relative importance of this variability in the total signal
variability. As a result, the energy distribution between wavelet details for each borehole in the database can be extracted and
mapped.

Then, we used the Continuous Wavelet Transform (CWT) for visualizing the spectral content of GWL time series of
representative boreholes of each GWL variation type in the Paris Basin. CWT is a widely used method for identifying scales
of variability in environmental time series (Torrence and Compo, 1998; Labat, 2005; Liesch and Wunsch, 2019). The
litterature about CWT is very rich and theoretical background along with an application to climatic variables are available in
Torrence and Compo (1998). The CWT produces a time-scale (or time-period) contour diagram on which time is indicated
on the x-axis, period or scale on the y-axis and amplitude (or variance, or power) on the z-axis.





These analyses used R packages wmtsa (Constantine and Percival, 2016) and biwavelet (Gouhier and Grinsted, 2012).
**3.2. Influence of low-frequency on the occurrence of high and low groundwater levels**
The influence of groundwater LFV (multi-annual and decadal) on HL and LL was estimated with the MODWT method for
the 78 selected boreholes. As seen in the 3.1. section, summing up all wavelet details and the last smooth rebuild the original
signal. Based on this assessment, we subtracted the interest wavelet detail, corresponding to a specific timescale, from the
original signal to evaluate its influence on HL and LL (Fig. 2a). This method allowed us to assess whether the withdrawal of
multi-annual (~7-yr) and/or decadal (~17-yr) components leads to a different number and level of GWE in the filtered
groundwater time series compared to the original series.

First, HL and LL were identified in original groundwater time series when they exceed thresholds set at the percentile 0.8
and 0.2, respectively (Fig. 2a). Such percentiles were selected because they are the optimal thresholds to have a correct and
sufficient number of HL and LL on a 44 years period, particularly for time series with inertial GWL variation type. Once
these HL or LL have been identified, for each studied time series, details corresponding to multi-annual variability, decadal
variability, and both variabilities were successively subtracted from the original signal. From these filtered time series, we
evaluated if the subtraction of one given component affected HL peaks or LL peaks as initially identified in the original data.
In the case where HL peaks still exceeded the initial threshold, then the subtracted component(s) had little influence on HL
emergence. On the opposite, if peaks moved below the initial threshold, then the subtracted component(s) had significant
influence on HL emergence. The same assessment was realised with LL peaks.

For sake of clarity, in this paper the terms "extremes" or "extreme levels" refer to HL above percentile 0.8 and LL below
percentile 0.2.

For each borehole, we calculated a percentage describing the proportion of HL or LL in GWL generated by the considered
component(s) (i.e., HL or LL that were no longer considered as such if the component was filtered). This calculation is
presented below with HL as instance:






$$Percentage\ of\ HL\ generated\ by\ the\ component = \left(\frac{number\ of\ HL\ moving\ below\ threshold\ in\ filtered\ signal}{number\ of\ HL\ in\ original\ signal}\right) * 100 \ (1)$$

Then, these results were mapped for each borehole and each filtered component (multi-annual, decadal or both).

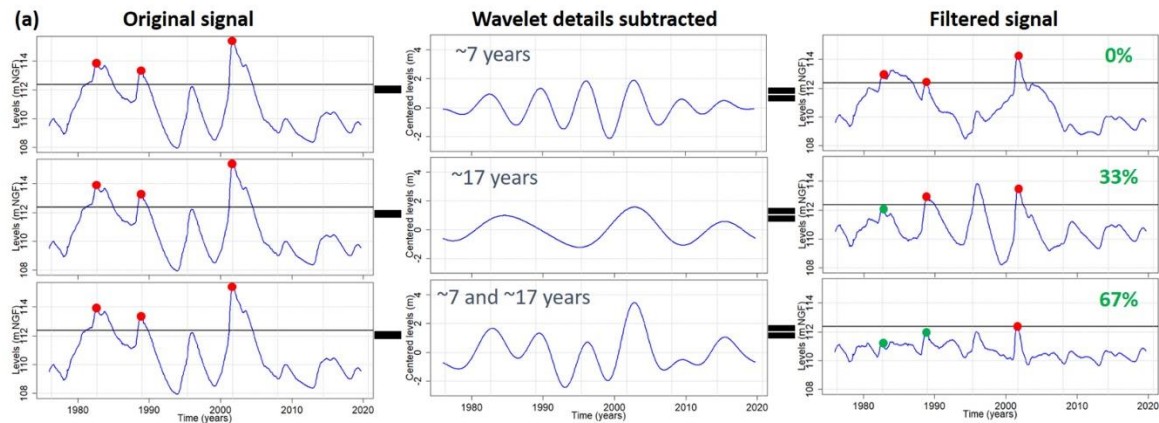

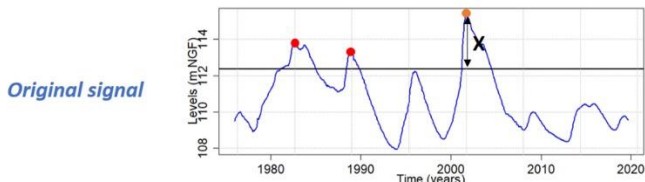

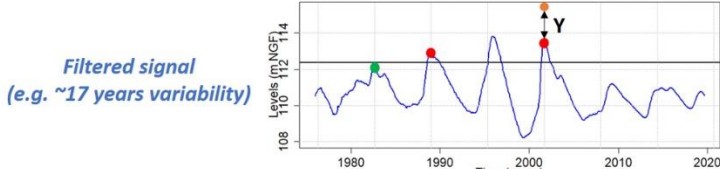

$$Contribution\ in\ the\ amplitude\ of\ threshold\ exceedance = \left(\frac{Y}{X}\right) * 100 \quad (2)$$


**Figure 2: Workflow of (a) the influence of low-frequency variability on high and low groundwater levels occurrence (example of**
**high levels); (b) the contribution of low-frequency variability in the amplitude of threshold exceedance. The borehole of**
**Goupillières (chalk of Normandy) is taken as an example.**



### 3.3. Role of groundwater low-frequency variability in the amplitude of threshold exceedance

The calculation of the contribution of groundwater LFV in the amplitude of threshold exceedance (ATE) was achieved with the MODWT method. The methodology consisted in filtering the wavelet details of interest (corresponding to multi-annual and decadal variabilities) to estimate how they impact the amplitude of HL or LL peaks (Fig. 2b).

First, we estimated the difference between the reached level and the threshold value in the original signal (Fig. 2b; Step 1). Then, we estimated the difference between the reached level in the original signal and the obtained level after filtering the interest detail(s) (Fig. 2b; Step 2). This difference revealed the amplitude of levels carried by the subtracted detail(s). Finally, we estimated the contribution of the filtered component in the amplitude of threshold exceedance with the following equation (Fig. 2b; Step 3):

$$Percentage\ of\ contribution = \left(\frac{Y}{X}\right) * 100 \quad (2)$$

Where $Y$ represented the difference between the observed real level and the obtained level after filtering; $X$ represented the difference between the observed real level and the threshold value.

From this calculation, 3 types of contribution of multi-annual and/or decadal components were highlighted:

- In case of negative percentage, we observed an attenuation of the HL or LL peak owing to the presence of the component considered, meaning that the level reached would have been higher than actually observed without attenuation by this component.
- In case of positive percentage lower than 100%, we observed an amplification of the HL or LL peak by the component, meaning that without this component the reached level was lower (HL) or higher (LL) than the actually reached level but still above (HL) or below (LL) the threshold and the HL or LL remained an extreme.
- In case of positive percentage higher than 100%, HL or LL peak was generated by the component, meaning that without this component the reached level fall below (HL) or above (LL) the threshold and the HL/LL was no longer considered as an extreme.





This analysis allowed us to better understand the contribution of the LFV in the emergence of HL and LL and estimate its
contribution in HL/LL amplitude. It was conducted on major HL/LL events of Paris Basin: 1995 and 2001 for HL, 1992 and
1998 for LL. We focused precisely on these four events because they are currently among the most severe hydrogeological
droughts and floods events across the Paris Basin (Deneux and Martin, 2001; Machard de Gramont and Mardhel, 2006;
Seguin et al., 2019). Knowing that the establishment of GWL droughts may occur a few years apart between two boreholes
(even in the same hydrogeological entity), we detected the LL peaks on a window extending of more or less 3 years before
and after the 1992 and 1998 events to be sure to correctly consider the lowest level.
**4.    Multi-timescale variability of groundwater levels in aquifers of the Paris Basin**
Across the Paris Basin, various types of GWL variation were highlighted depending on the hydrogeological entity
considered, according to the dominant time-scale components which characterize their variability (Fig. 3). The Beauce
limestones (entity H) consist the most inertial system amongst entities of Paris Basin with a large predominance of decadal
variability (DV; purple in Fig. 3). This DV is also significant and even predominant in southern Lutetian and Ypresian sands
of Paris Basin (F) and the southern Seno-Turonian chalk of Normandy (B). Farthest north in these two hydrogeological
entities, the importance of the DV shrinks to be present in almost equal proportion with the multi-annual variability (MAV;
darkblue in Fig. 3). The MAV becomes predominant in the northern Seno-Turonian chalk of Normandy/Picardy (B). From
eastern Seno-Turonian chalk of Normandy/Picardy (B) to northern Seno-Turonian chalk of Artois-Picardy (C), the MAV
constitutes a half to a quarter of total variability, while the DV diminishes significantly in the Artois-Picardy basin.

On the opposite, the annual variability (AV; pink in Fig. 3) swells from eastern Seno-Turonian chalk of Normandy/Picardy
to Artois-Picardy to represent up to a quarter of total groundwater variability (Fig. 3). The AV is also significant, and even
predominant compared to MAV and DV, in GWL of Champagne and Bourgogne chalk (D and E). These entities exhibit the
most reactive water tables in our study.



Finally, the Jurassic limestones of Bessin (A) exhibit two types of GWL variation: inertial farthest south with predominant
MAV and DV, and more reactive on the border of English Channel with AV and inter-annual variability (light blue in Fig. 3)
accounting for a half of total variability.

No typical GWL variation was identified in the Eocene limestones of Paris Basin (G; Fig. 3).

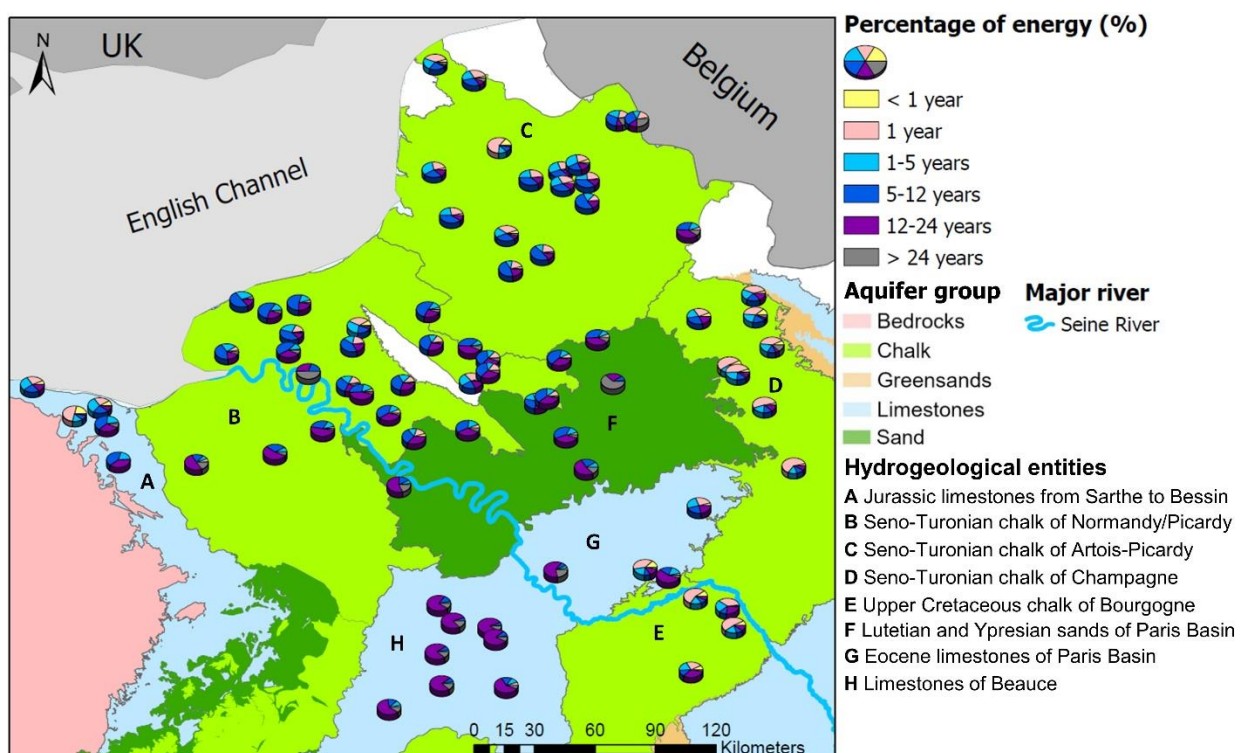

**Figure 3: Multi-timescale variability of groundwater levels in Paris Basin (78 boreholes). Pie charts describe the energy percentage**
**of each timescale of variability reflecting their importance in total groundwater level variations.**
Overall, hydrogeological entities in the Paris Basin display 4 major types of GWL variation:
• Type iD: inertial with a predominant Decadal variability such as entity H (Fig. 3 and 4)
• Type iMD: inertial with predominant Multi-annual and Decadal variabilities such as entities B, F and southern part
of entity A (Fig. 3 and 4)





- Type cAM: combined with predominant Annual and Multi-annual variabilities such as entity C (Fig. 3 and 4)

- Type cA: combined with a predominant Annual variability such as entities D and E (Fig. 3 and 4)

**Figure 4: Time series (bottom) and wavelet spectra (up) of a typical time series representing each major groundwater level variation type. For type iD, this is the time series of Thionville borehole (Beauce limestones – entity H); for type iMD, Blacqueville borehole (chalk of Normandy – entity B); for type cAM, Pihen-Lès-Guînes borehole (chalk of Artois-Picardy – entity C); and for type cA, Grandes-Loges borehole (chalk of Champagne – entity D).**





## 5. Influence of low-frequency variability on the occurrence of high and low groundwater levels

**5.1. Spatial distribution across the Paris Basin**

This section aims determining to what extent the LFV influences the quantity of HL and LL in groundwater time series over the 1976-2019 period and what percentage amongst these extreme levels were generated by the MAV (~7-yr), the DV (~17-yr) and the combination of both.

Figure 5 displays for each groundwater time series the number of HL peaks above a threshold set on the percentile 0.8 and LL peaks below a threshold set on the percentile 0.2 through hydrogeological entities of Paris Basin. Types iD and iMD entities displayed the lowest number of HL and LL peaks with a decadal occurrence (H) or a multi-annual occurrence (B, F, southern A). Conversely, the quantity of HL and LL increased significantly in types cAM and cA entities with a quasi-annual occurrence throughout multi-annual HL and multi-annual LL, respectively (entities C, D, E, and northern A). These results highlighted the significant control of the LFV on the number of HL and LL peaks: the more the LFV is predominant in GWL, the more the number of peaks is reduced because they are supported by this LFV, which naturally contains a few extremes over a relatively short period of only a few decades.





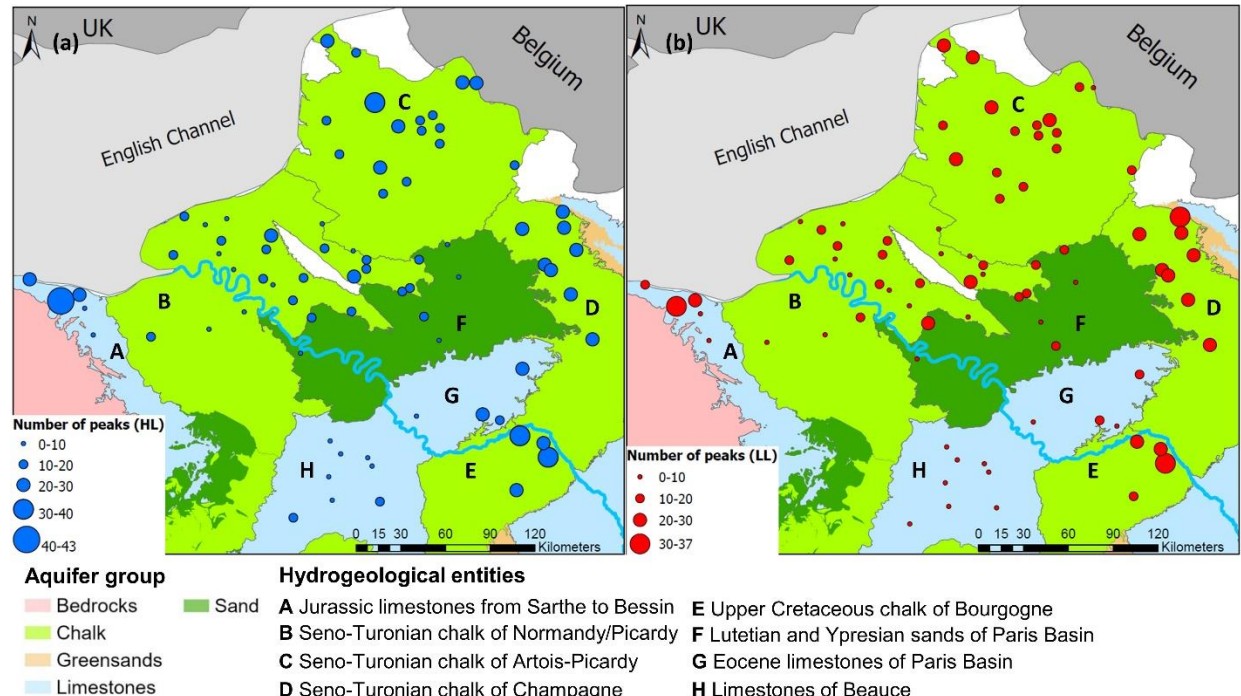

**Figure 5: Number of HL peaks above percentile 0.8 (a); and LL peaks below percentile 0.2 (b) over the 1976-2019 period.**

Amongst these HL and LL peaks, we estimated for each groundwater timeseries what percentage of HL and LL are generated by MAV, DV and both variabilities (Fig. 6). The percentage of HL and LL generated by a given variability was closely related to the GWL variation type in hydrogeological entities.





**Percentage of high/low levels**
**generated by the variability**

0%    50%    100%

**Aquifer group**

- Bedrocks
- Chalk
- Greensands
- Limestones
- Sand

**Hydrogeological entities**

**A** Jurassic limestones from Sarthe to Bessin
**B** Seno-Turonian chalk of Normandy/Picardy
**C** Seno-Turonian chalk of Artois-Picardy
**D** Seno-Turonian chalk of Champagne
**E** Upper Cretaceous chalk of Bourgogne
**F** Lutetian and Ypresian sands of Paris Basin
**G** Eocene limestones of Paris Basin
**H** Limestones of Beauce






**Figure 6: Percentage of HL or LL generated by the ~7-yr (MAV), ~17-yr (DV), ~7-yr and ~17-yr components in hydrogeological**
**entities of Paris Basin. This percentage corresponds to the proportion of HL or LL that are not considered as extreme levels (being**
**respectively below or above threshold) when the component(s) is (are) absent from the original signal, meaning that these HL or**
**LL are significantly supported by the component.**

For the type iD entity (H), the DV generated 100% of HL and LL. Exceptions were noticeable for the two southernmost
boreholes that might be related to the weaker significance of the DV in GWL.

For type iMD entities (B, F, southern A), the LFV had a lesser influence on both HL and LL. The combination of both
variabilities (MAV and DV) explained the emergence of at least 50% of LL. This was also often the case for HL but in lesser
proportions. Individually, the MAV and DV still explained a rather large proportion of HL and LL.

For the type cAM entity (C), the influence of LFVs on GWE was reduced compared to types iD or iMD entities, particularly
on HL. A larger proportion of LL than HL was influenced by the LFV. Indeed, less than 50% of HL were generated by the
combination of MAV and DV. Conversely, more than 50% of LL were generated by the combination of MAV and DV in
southern C. In the northern part, the proportion approached the 50% although it did not exceed it. This significant proportion
of LL generated by the combination of both variabilities seemed to be directly related to the influence of the MAV.

For type cA entities (D and E), the proportion of HL and LL generated by the LFVs was relatively small. Individually, the
MAV and DV explained the emergence of less than 50% of HL and LL identified in the time series. The combination of
both variabilities did not allow either to explain the emergence of more than 50% of HL and LL.

Overall, the LFVs seemed to have a larger influence on LL than HL.





### 5.2. One century of high and low groundwater levels and low-frequency variations

The time series used previously for assessing the spatial distribution of high and low GWL as controlled by LFV are rather short, banning any assessment of long-term variations of high- and low GWL. To explore it, the same analysis than in section 5.1. was conducted for the borehole of Tincques monitoring the Seno-Turonian chalk of Artois-Picardy (entity C) and providing data since 1903 (Fig. 7 and 8).

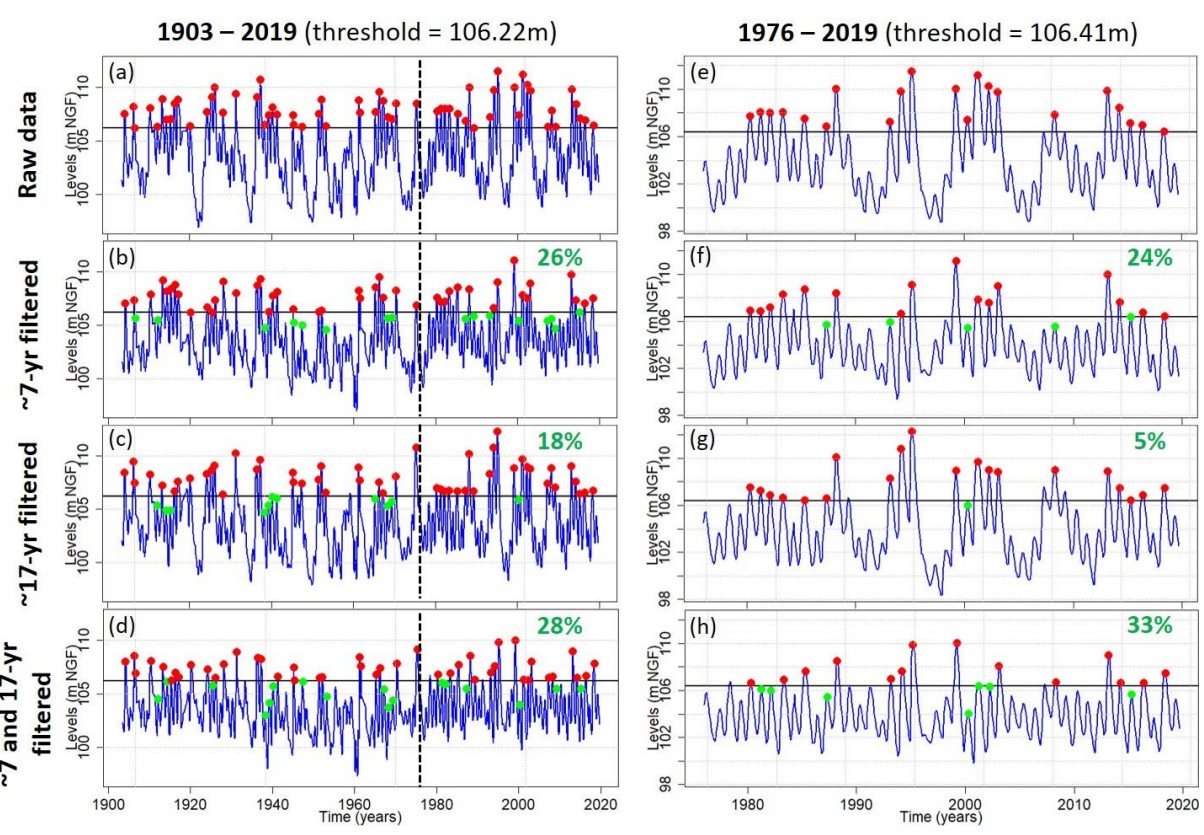

**Figure 7: Influence of ~7-yr (MAV), ~17-yr (DV), ~7-yr and ~17-yr variabilities on the occurrence of HL for the Tincques' GWL over the 1903-2019 and 1976-2019 periods. The percentage of HL generated by the filtered component(s) (i.e., moving below the percentile 0.8) is indicated in green. The dotted black line represents the 1976 year.**

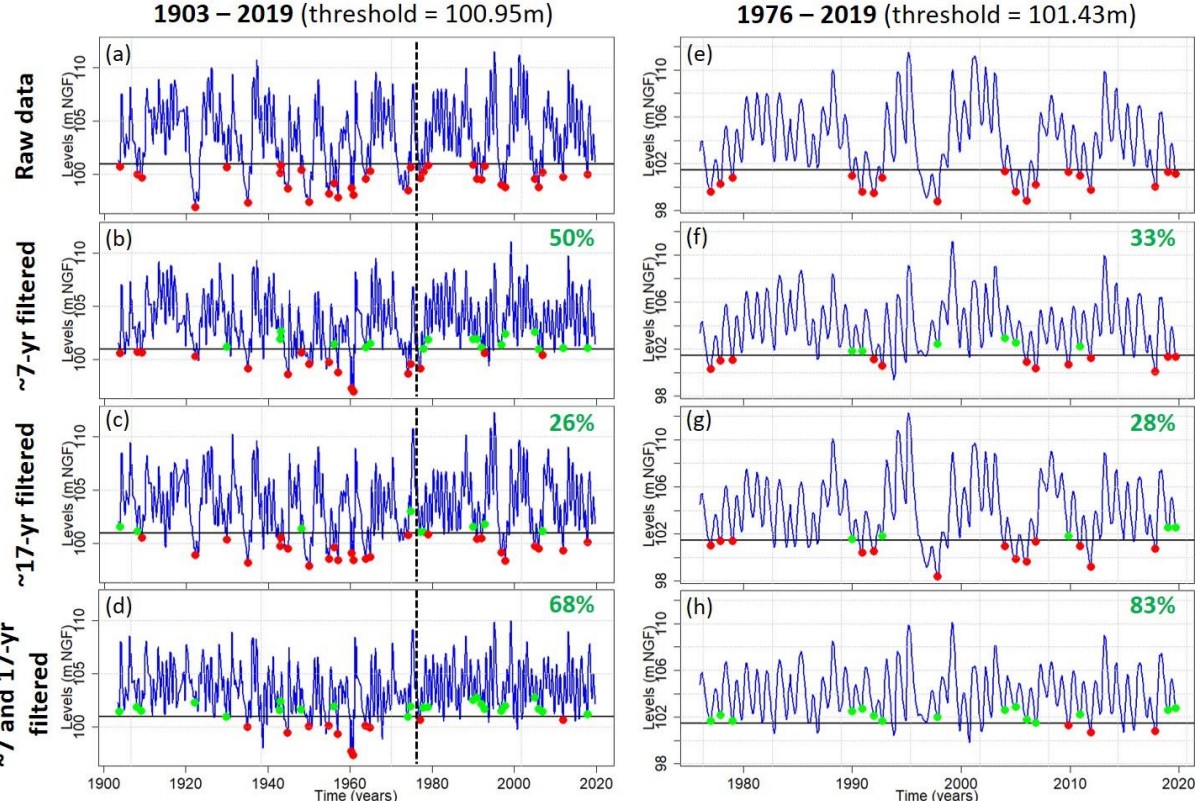

**Figure 8: Influence of ~7-yr (MAV), ~17-yr (DV), ~7-yr and ~17-yr variabilities on the occurrence of LL for the Tincques' GWL over the 1903-2019 and 1976-2019 periods. The percentage of LL generated by the filtered component(s) (i.e., moving above the percentile 0.2) is indicated in green. The dotted black line represents the 1976 year.**

The predominant components of GWL variability were extracted and plotted on figure 9 for both 1976-2019 and 1903-2019 periods. On the long-term (1903-2019), the combination of MAV and DV explained ~50% of GWL total variance, that was consistent with the percentage obtained on the short term for Tincques time series but also for the others GWL time series in the South part of entity C (Fig. 9 and 3).



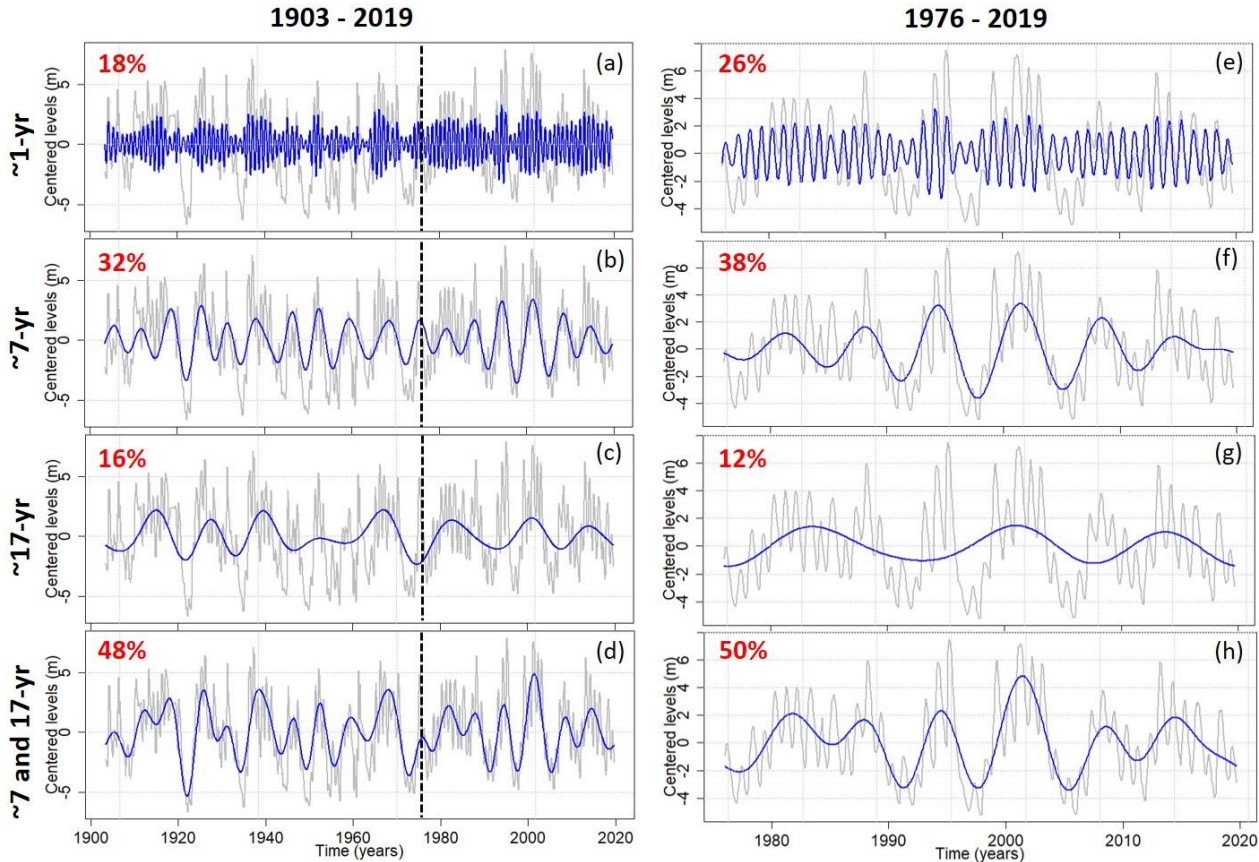

**Figure 9: Maximum overlap discrete wavelet transform (modwt) of groundwater levels at Tincques in the Seno-Turonian chalk of Artois-Picardy over the 1903-2019 and 1976-2019 periods. In grey is displayed the original time series, in blue the wavelet detail, in red the energy percentage.**

First, the identified thresholds (i.e., percentiles 0.8 and 0.2) were lower on the longer period (1903-2019) than the shorter one (1976-2019) indicating lower GWL in average and more severe hydrogeological droughts before the 1970s (Fig. 7 and 8).

The proportions of HL and LL generated by the MAV, DV, and the combination of MAV and DV on both periods for Tincques (1976-2019 and 1903-2019) were compliant to the ranges of percentages exhibited by boreholes monitoring the Seno-Turonian chalk of Artois-Picardy over the 1976-2019 period (Fig. 7, 8 and 6). Similarly to the previous conclusions, we also observed a higher influence of LFVs on LL than HL (Fig. 7 and 8).






Some HL that fell below threshold when the MAV or DV was filtered out from the original signal – and then being no
longer considered as GWE – did not fall below the threshold when both MAV and DV were filtered out, remaining GWE
(Fig. 7). As instance, we can observe such phenomenon for the 3rd HL on the 1903-2019 period (Fig. 7b and 7d). This is
related to the compensation between both MAV and DV: the withdrawal of DV increased the level compared to raw data
(Fig. 7c and 7a), much more than the level decreased when the MAV was filtered out (Fig. 7b and 7a), therefore the level
stayed above the threshold when both variabilities were filtered out (Fig. 7d). The same phenomenon is visible for some LL
as well (Fig. 8).

Proportions of HL generated by the MAV (Fig. 7b and 7f) or the combination of MAV and DV (Fig. 7d and 7h) were close
between both periods (1976-2019 and 1903-2019). Over the 1903-2019 period, the HL generated by the MAV (Fig. 7b) or
the combination of MAV and DV (Fig. 7d) were regularly distributed over time. Conversely, the proportion of HL generated
by the DV was much higher for the 1903-2019 period (Fig. 7c) than the 1976-2019 period (Fig. 7g), the DV having
seemingly much more influence on HL before the 1980's.

LL peaks were found to be much more pronounced before the 1960's (Fig. 8a). The lowest level in 1921 was significantly
supported by the combination of MAV and DV (Fig. 8d and 9d). The LL between ~1930 and ~1970 remained present even
when both MAV and DV were removed (Fig. 8d). A multi-decadal LL period between ~1930 and ~1970, supporting the LL
peaks, explained why the proportion of LL generated by the combination of both variabilities on the 1903-2019 period (Fig.
8d) was much lower than the one displayed on the 1976-2019 period (Fig. 8h). The DV generated a similar proportion of LL
for both studied periods (Fig. 8c and 8g). Finally, the proportion of LL generated by the MAV was larger over the 1903-2019
period (Fig. 8b) than the 1976-2019 period (Fig. 8f), with the majority of LL generated by this variability emerging since the
1960's.





**6. Contribution of groundwater low-frequency variability to the emergence of well-know historical events**

The following subsections aim determining the contribution of either MAV or DV or both of them in the emergence of

historical events: 1995 and 2001 for HL, 1992 and 1998 for LL (Fig. 10 and 11). This study was based on a percentage of

contribution of the ~7-yr (MAV), ~17-yr (DV) or ~7 and ~17-yr components in the ATE. Three results can be reached: an

attenuation of the HL or LL peak by the component when the percentage is <0% (i.e., the level was higher (HL) or lower

(LL) than original level when the component was filtered); an amplification of the HL or LL peak by the component when

the percentage is between 0% and 100% (i.e., the level was lower (HL) or higher (LL) than original level when the

component was filtered but staying above (HL) or below (LL) the threshold and remaining an extreme); the generation of the

HL or LL peak by the component when the percentage is ≥100% (i.e., the level was no longer considered as an extreme

when the component was filtered moving below (HL) or above (LL) the threshold).

In addition, figure 12a displays for 3 boreholes representative of each GWL variation type of chalk aquifers (type iMD –

Blacqueville, type cAM – Pihen-Lès-Guînes, type cA – Grande-Loges) the modwt extraction of AV (~1-yr), MAV (~7-yr),

DV (~17-yr), MAV and DV (~7-yr and ~17-yr) with the studied historical events highlighted. It provides a visual insight of

the situation (i.e., positive or negative level) of these variabilities during the emergence of historical events. The same

extraction is also realised in EP to be compared to GWL (Fig. 12b).

In the subsequent section, the term "concomitant situation" refers to concomitant minima or concomitant maxima levels of

MAV and DV. The term "opposite situation" refers to maxima-minima or minima-maxima levels of MAV and DV.

**Figure 10: Contribution of ~7-yr (MAV), ~17-yr (DV), ~7-yr and ~17-yr components in the amplitude of threshold exceedance (ATE): case of 1995 and 2001 high levels (percentile 0.8). It indicates if the component generates (contribution in ATE ≥ 100%), attenuates (contribution in ATE < 0%) or amplifies (contribution in ATE between 0% and 100%) the high level. In case of missing borehole(s) on maps, it means that either the 1995 or 2001 high level was not identified in the time series (i.e. no threshold exceedance).**


461


**Figure 11: Contribution of ~7-yr (MAV), ~17-yr (DV), ~7-yr and ~17-yr components in the amplitude of threshold exceedance (ATE): case of 1992 and 1998 low levels (percentile 0.2). It indicates if the component generates (contribution in ATE ≥ 100%), attenuates (contribution in ATE < 0%) or amplifies (contribution in ATE between 0% and 100%) the low level. In case of missing borehole(s) on maps, it means that either the 1992 or 1998 low level was not identified in the time series (i.e. no threshold exceedance).**



**Figure 12: Extraction of ~1-yr (AV), ~7-yr (MAV), and ~17-yr (DV) components in (a) groundwater levels of three boreholes monitoring chalk aquifers and in (b) effective precipitation corresponding to the three boreholes. The 1992 low level is highlighted in grey, the 1995 high level in orange, the 1998 low level in yellow, the 2001 high level in purple. The energy percentage of each component or association of components is indicated in red.**



### 6.1. The low-frequency origin of the selected historical events

The low-frequency origin of each historical event was spatially consistent across aquifers of the Paris Basin (Fig. 10 and 11).

First, the 2001 HL and the 1992 LL resulted of a concomitant situation of MAV (~7-yr) and DV (~17-yr), both leading to the

accentuation of the amplitude of the extreme level observed. Figure 12a also highlights such situations for both events via

the MODWT analysis of GWL for three boreholes in the chalk. Conversely, the 1995 HL and the 1998 LL resulted of an

opposite situation of the MAV and DV, leading to the attenuation of the extreme level (Fig. 10 and 11). Indeed, the 1995 HL

originated from a multi-annual HL attenuated by a decadal LL, while the 1998 drought originated from a multi-annual LL

attenuated by a decadal HL (Fig. 12a).

We also found such concomitant or opposite situations of MAV and DV in Effective Precipitation (Fig. 12b). The presence

of such variabilities in EP indicates their climatic origin. Cross-correlation in figure 13 indicated that such LFVs in GWL

lagged those in EP from 0 month for the most reactive system (chalk of Champagne – Type cA) to 1.3 years for the most

inertial system (Seno-Turonian chalk of Normandy – Type iMD).

Although such concomitant or opposite situations of MAV and DV in GWL were overall consistent within and between the

hydrogeological entities, some discrepancies can be highlighted locally. The influence of the DV during the 1998 drought is

a typical example: while it attenuated LL, it also sporadically amplified LL in some places (entities A, B, C, D, F, G and H in

Fig. 11). Such discrepancies might be explained by local basin properties that may operate different filtering effects. This

phenomenon was particularly recuring in GWL of entity H, and this is developed in section 6.2.1.

In addition, the contribution of each component to the ATE differed according to hydrogeological entities and the GWL

variation type (Fig. 10 and 11). Therefore, the subsequent section aims assessing the specific contribution of MAV and DV

to explain the amplitude of HL and LL for each hydrogeological entity.



Figure 13: Multi-annual (~7-yr - MAV) and decadal (~17-yr - DV) components of effective precipitation (black) and groundwater

levels (red) in chalk aquifers of Paris Basin (left side); Cross-correlation between MAV and DV components of effective

precipitation and groundwater levels (right side).





## 6.2. Contribution of low-frequency variabilities to amplitude of threshold exceedance

### 6.2.1. Type iD (inertial – decadal dominant) hydrogeological entities

Entity H displayed the largest contribution of the LFV (i.e., the combination of MAV and DV) in the HL and LL emergence (Fig. 10 and 11). In such typical behaviour with a large predominance of the DV in GWL, the LFV was involved for at least 100% in the ATE and always generated the HL or LL, regardless the event. This contribution is primarily related to the DV that alone is responsible for at least 75% of the ATE. During the 1998 LL, the DV in entity H displayed an opposite influence to that of other hydrogeological entities (Fig. 11). While the DV primarily attenuated the LL peak for the other entities, it generated the LL peak in entity H. Indeed due to its large predominance in GWL, the DV can never attenuate LL (or HL) peaks because this component supports exclusively the HL and LL peaks.

### 6.2.2. Type iMD (inertial – multi-annual and decadal dominant) hydrogeological entities

Such entities (B, F, southern A) exhibited a significant contribution of the LFV (i.e., the combination of MAV and DV) in the HL and LL emergence (Fig. 10 and 11). During the 2001 HL, the LFV was involved for at least 50% in the ATE and this originated from MAV and DV which were both involved in rather similar proportions (Fig. 10). During the 1995 HL, the contribution of the LFV was more reduced due to the attenuation of the HL peak by the DV, and the threshold exceedance was related primarily to the MAV accounting for at least 50% of the ATE (Fig. 10).

During the drought events, the contribution of the LFV was enhanced as it generated the LL events (Fig. 11). For the 1992 drought, the MAV and DV individually accounted for at least 50% in the ATE and may even have generated the LL peak. While this event was generated primarily by the combination of a multi-annual and a decadal LL, conversely the 1998 drought was generated by the MAV alone.

For the four historical events, this is primarily the MAV that drove GWL and guided the reach of a LL or a HL (Fig. 12a; Blacqueville). Due to its large amplitude in GWL over the 1990-2010 period, the highest or lowest levels (over 1976-2019) were reached often during this period. Simultaneously, the DV amplified or attenuated the reached HL or LL. The significance of this attenuation or amplification is directly controlled by the importance of DV in GWL but also by its phase.





Hence, the greater the component accounts for a significant part in total GWL variability, the greater the attenuation or amplification. The potential dephasing and distortion of the component induced by the physical and morphometric properties of catchments may also influence on the significance of the attenuation or amplification. The more the HL on the top of decadal HL or the LL in the concavity of decadal LL, the more the amplification. And the more the LL on the top of decadal HL or the HL in the concavity of decadal LL, the more the attenuation.

The capability of these hydrogeological entities to exhibit a significant DV in GWL, that is responsible for the amplification or the attenuation of HL and LL, induced that the lowest and the highest levels (in raw data) were not necessarily reached during the lowest and highest multi-annual levels (Fig. 12a; Blacqueville). For instance, the lowest level in the raw data was not reached in 1998 when the MAV exhibited its lowest level since the DV in positive phase attenuated this low level, but it was reached in 1992 when the low multi-annual level was less severe but accentuated by the low decadal level. Therefore in such systems, the severity of droughts and HL depends on both MAV and DV.

### 6.2.3. Type cAM (combined – annual and multi-annual dominant) hydrogeological entities

For the entity C, the LFV (i.e., the combination of MAV and DV) was significantly involved in the LL emergence, while it was more weakly involved in the HL emergence (Fig. 10 and 11). The 1995 HL exhibited the lowest contribution of the LFV due to the opposite situation of MAV and DV (multiannual HL vs decadal LL; Fig. 10). Consequently, LFV was involved primarily for less than 50% in the ATE. Conversely, due to the concomitant maxima of MAV and DV during the 2001 HL, the contribution of LFV was enhanced accounting for 25% to 75% in the ATE (Fig. 10). During drought events, the LFV was much more involved accounting for at least 75% in the ATE and may even have generated the LL (Fig. 11).

The DV being poorly significant in GWL of the entity C, it was therefore consistent to observe a weak contribution of this variability in the emergence of the 1992 LL and the 2001 HL (Fig. 10 and 11). Conversely, the MAV was more involved because this is the predominant LFV in GWL. In the southern C, the MAV alone generated both LL events (Fig. 11).





The weak DV in GWL did not allow HL and LL to be significantly attenuated or accentuated. In the entity C, the amplitude
of a LL or HL was directly dependent on the amplitude of the MAV (without considering the AV) (Fig. 12a; Pihen-Lès-
Guînes). For instance, the lowest level in raw data occured in 1998 when the MAV exhibited its lowest level, and the 1992
LL was less severe like in the MAV. Therefore in such systems, the severity of droughts are almost only dependent on the
MAV. In contrast, the severity of HL depends on both MAV and AV.
**6.2.4.     Type cA (combined – annual dominant) hydrogeological entities**
Entities D and E generally displayed the slightest contribution of the LFVs in the HL and LL emergence (Fig. 10 and 11).

Generally, the contribution of the LFV (i.e., the combination of MAV and DV) in the ATE remained lower than 100% for
the four events. Locally, this contribution was higher than 100% and generated the HL or LL. However, it remained rather
rare at the scale of these two hydrogeological entities. The highest contributions of the LFV in the ATE were found during
events displaying concomitant situation of the MAV and DV (Fig. 10, 11 and 12a – Grandes-Loges). Indeed during the 2001
HL and the 1992 LL, the LFV explained at least 50% of the ATE. Conversely, the LFV explained less than 50% of the ATE
for events displaying an opposite situation of MAV and DV (1995 HL and 1998 LL).

Individually, the MAV was involved for less than 100% in the ATE of these four events (Fig. 10 and 11). This contribution
fluctuated significantly across the entities from 0% to 100% and according the event. During the 2001 HL, the contribution
of the MAV was larger than that of the DV. Conversely during the 1992 LL, the respective contribution of the MAV and DV
was rather similar.

Compared to type iD, iMD and cAM entities, differences between the contribution of the LFV in ATE during HL and LL
events are less striking (Fig. 10 and 11). In addition, due to the quasi-equal energy distribution between MAV and DV (even
if they remain rather weak in total variability), the severity of LL and HL is dependent on the amplitude of both MAV and
DV (Fig. 12a; Grandes-Loges). The MAV primarily guided the emergence of a HL or a LL, while the DV accentuated or





attenuated these HL or LL. In addition, the predominant AV can even more accentuate or attenuate the HL severity but also
that of LL.

## 7. Discussion

The present study showed the large influence of the MAV on groundwater LL occurrence in types iMD, cAM, cA aquifers.
Simultaneously, the DV modulates the severity of droughts in aquifers for which it accounts for a significant part of GWL
variability (types iMD or cA aquifers). Therefore in such contexts, the DV can significantly mitigate LL or amplify LL
amplitude. When the DV only accounts for a small part of total variability, then the severity of droughts mainly depends on
the amplitude of MAV. These observations are also valid for groundwater HL, but the influence (in proportion) of LFV on
HL is mitigated compared to LL since the amplitude of high-frequency variabilities (infra-annual to annual variabilities)
strenghen during wet periods (Fig. 12).

These LFVs, modulated in variance by catchments properties, are directly linked to internal climate variability
(Gudmundsson et al., 2011). Numerous studies highlighted the potential link between the LFVs in hydroclimatic variables
over the European continent and the LFVs of well-known climatic or oceanic modes such as the North Atlantic Oscillation
(NAO) or the Atlantic Multidecadal Oscillation (AMO) also known as the Atlantic Multidecadal Variability (AMV) (Massei
et al., 2010; Boé and Habets, 2014; Neves et al., 2019; Liesch and Wunsch, 2019). Such links have even been very well
documented and already very extensively studied for the northern France area in several studies dating back the end of the
2000s (Massei et al., 2007; Slimani et al., 2009; El Janyani et al., 2012; Fritier et al., 2012; Massei and Fournier, 2012;
Massei et al., 2017).

In the past, regular changes of hydrological variability (i.e., variance) have been observed at each timescale in numerous
studies (Fritier et al., 2012; Dieppois et al., 2013 and 2016; Massei et al., 2010 and 2017; Neves et al., 2019). Knowing the
dependence of GWE to LFVs, this aperiodic behaviour can heavily influence the HL and LL severity in aquifers displaying
inertial or combined GWL variation types. This is why, we found in our study varying contributions of MAV and DV in





GWE emergence. Indeed, there are periods where LFVs can exhibit an attenuated variance (e.g., since the end of 2000's for
the ~7-yr variability; Fig. 9f) or on the contrary an accentuated variance (e.g., 1990s to 2000s for the ~7-yr variability; Fig.
9f). Hence, HL and LL happening during periods with an increased variance of LFV are generally much more severe than
those happening during periods with attenuated variance. Therefore, this aperiodic behaviour of LFVs considerably limits
the predictability of groundwater droughts, or more largely of GWE.

The identification of large-scale atmospheric and oceanic states leading to variance modifications in streamflow,
precipitation, groundwater time series is still a major issue. Recently, Haslinger et al. (2021) highlighted that the increase of
precipitation variability across the Alps at the interannual timescale can be related to a predominant meridional circulation
(linked to a positive SST anomaly gradient) enhancing soil moistures feedbacks, while the decrease of variability can be
related to a predominant zonal circulation (linked to a negative meridional SST anomaly gradient) suppressing soil moistures
feedbacks. In addition, they underlined that the increase variability of precipitation occurs during AMV positive phases at a
~50-yr timescale. Knowing the large impact of LFV amplitude changes on extreme levels, particularly on GWL due to the
low-pass filter effect of aquifers, such studies should be developed to identify the drivers of variability changes for predictive
purposes of extreme levels.

Although the potential link between hydrological variability and climate variability is rather well-established, the long-term
forecasting of large-scale variabilities (i.e., multiannual, decadal) remains complex owing to their stochastic nature. Indeed,
albeit what is seemingly claimed in some studies, none of these variabilities can be considered periodic, and this is also the
case of those constituting the NAO spectrum (e.g., Fernandez et al., 2003) as well as the subsequent variabilities observed in
hydrology. This issue constrains the robustness of climate projections, in which the internal climate variability is often
poorly reproduced and appears as a major source of uncertainty (Qasmi et al., 2017). For instance, Terray and Boé (2013)
estimated that uncertainties related to internal variability in precipitation projections over France in the middle of the 21st
century may be as large as uncertainties due to climate models. Therefore, such uncertainties in precipitation estimation may
also have a huge impact on the future estimation of aquifers recharge and in fine on the predictability of GWL and extremes.






It must also be underlined that anthropogenic forcing may have already impacted climate variability (Lenton et al., 2008;
Dong et al., 2011; Caesar et al., 2018). Moreover, Feliks et al. (2011) highlighted that interannual variabilities can be
suppressed from atmospheric circulation when they are omitted from the Gulfstream front or the SST field. Such
modifications of the LFV in large-scale predictors may then influence the hydrological variability. The questions raised by
such a phenomenon are: to what extent this could impact the LFV in GWL and then influence extreme levels? Would that
lead to more extreme levels or less extreme levels? Would that lead to more or to less severe extremes? Would this be
expressed in the same way in aquifers exhibiting different GWL variation types?

The analysis of a long-term time series shows the importance of historical hindsight, particularly to highlight the influence of
a multidecadal variability on GWE. This multidecadal variability displayed by the Tincques time series when filtering MAV
and DV is also highlighted in streamflows through metropolitan France (Boé and Habets, 2014; Bonnet et al., 2017 and
2020). The links between this variability in streamflow and the AMV have been robustly established by Boé and Habets
(2014), Bonnet et al. (2020), and previously in spring precipitation by Sutton and Dong (2012). The negative phase of
multidecadal variability in streamflows and precipitation over the 1940-1960 period (corresponding to a warm period of the
AMV) is also exhibited in GWL of the Artois-Picardy chalk, hindering the ATE of LL to be fully explained by the MAV and
DV (Fig. 14). Low GWL are thus supported by a multidecadal variability seemingly associated to the multidecadal
variability in AMV. The short-term analysis could not highlight the significant contribution of multidecadal variability in
GWE emergence.






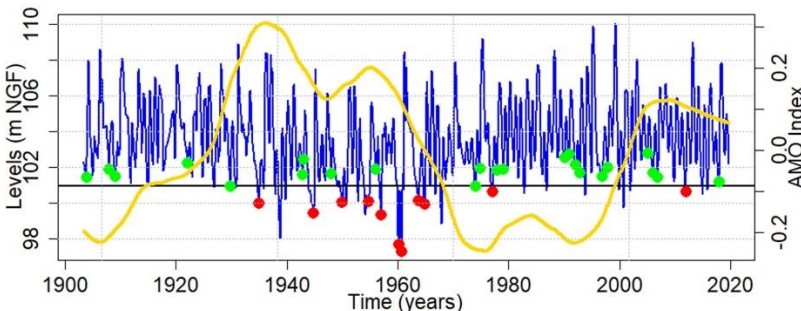


**Figure 14: Same as figure 8 but only for the Tincques' GWL filtered of both ~7-yr (MAV) and ~17-yr (DV) components, with the**

**Atlantic Multidecadal Variability (AMV) index superimposed in yellow. The AMV is smoothed to highlight the multidecadal**

**variability.**


The historical hindsight available for the Tincques time series also allowed us to observe tougher droughts before the 1960's,
certainly related to the negative phase of the multidecadal variability (Fig. 8a). Nonetheless, the most severe drought in the
Seno-Turonian chalk of Artois-Picardy appeared around 1921 (during a positive phase of the multidecadal variability). This
drought was also identified as one of the most severe events across Europe in precipitation and streamflow time series
(Folland et al., 2015; Caillouet et al., 2017; Rudd et al., 2017; Hanel et al., 2018; Barker et al., 2019). Bonnet et al. (2020)
also identified this event as the most severe hydrological drought in the Seine river flows. However in their study, this
drought did not appear as the most severe in the Beauce limestones (Toury time series), probably due to the high inertial
nature of this hydrogeological entity. Indeed in the chalk (Tincques time series), the 1921 drought seemed to be the result of
the combination between a low multi-annual level and a low decadal level (Fig. 9d). The combination of both components
explained almost exclusively the drought emergence. In the Beauce limestones, the MAV is poorly expressed in the GWL
signal, while the larger scale variabilities (decadal to multidecadal) are widely expressed, therefore the drought of 1921
could not have been as severe in the Beauce limestones as it was in the Seno-Turonian chalk, due to the significant role of
the MAV in its emergence. Nevertheless, the positive phase of the multidecadal variability probably mitigated the severity of
this drought in both aquifers.





The above example shows that a severe hydrogeological event in a given aquifer, supported by specific low-frequency
variabilities, may not be as severe in another aquifer when these variabilities are not or poorly significant in total GWL
variability. It highlights the resistance of aquifers to drought events for which the low-frequency components that support
them do not constitute a significant part of GWL variability.

Since these hydrogeological extremes are supported primarily by the LFVs in aquifers with inertial or combined GWL
variation types, and considering the large impact that they can induce on our societies (e.g., water exploitation, groundwater-
river exchanges, floods), it is necessary to identify the large-scale predictors of such variabilities for extreme levels
prediction purposes. In the litterature, it has been demonstrated that large-scale patterns leading to the different LFVs (i.e.,
multiannual, decadal) may vary across timescales (Massei et al., 2017; Sidibe et al., 2019). Such an approach should be also
conducted for GWL. In addition, the response of GWE to scenarios of changing climate variability (i.e., variance changes of
LFVs) should also be investigated.
**8.  Conclusion**
This study highlighted the heavy influence of the low-frequency variabilities (LFVs) on the occurrence of high and low
groundwater levels (GWL). First, we estimated the proportion of high levels (HL) and low levels (LL) among the total
number of their respective occurrences, that were generated by LFVs (typically, multi-annual and decadal variabilities) in
Paris Basin aquifers. These proportions were highly dependent on GWL variation types: for aquifers of type iD (inertial –
decadal dominant) and iMD (inertial – multi-annual and decadal dominant) occurrences of HL and LL were logically
strongly influenced by LFV, conversely to type cA (combined – annual dominant) and cAM (combined – annual and multi-
annual dominant) aquifers. In addition, the multidecadal variability also seemed to influence occurrences of HL and LL, but
it was only discernable on a 100-yr GWL time series.

Second, we determined the contribution of the LFVs to the amplitude of threshold exceedance (ATE) during four historical
events. Results highlighted that the contribution of the LFV was rather dependent on the significance of multi-annual and





decadal variabilities in the total GWL variability. This contribution varied according to the event. Generally, we also
observed a more significant contribution of the LFV in the ATE during LL compared to HL. This is related to the higher
contribution of high-frequency variability (infra-annual to annual) in HL events, since its variance increases during wet
periods, thus reducing the relative contribution of the LFV in the ATE of HL.

This study also highlighted that the occurrence of a HL or LL and its amplitude (or severity) seemed primarily guided by the
multi-annual variability in types iMD, cAM, and cA aquifers. The HL or LL severity can then be significantly accentuated
(e.g., 2001 HL, 1992 LL) or attenuated (e.g., 1995 HL, 1998 LL) by the decadal variability.

This study presented evidence about the key role of LFV in the occurrence of HL and LL. Since LFV originates from large-
scale stochastic climate variability as demonstrated in many previous studies in the Paris Basin or nearby regions, our results
point out that i) poor representation of LFV in GCM outputs used afterwards for developing hydrological projections can
result in strong uncertainty in the assessment of future hydrogeological extremes, ii) potential changes in the amplitude of
LFV, be they natural or induced by global climate change, may lead to substantial changes in the occurrence and severity of
hydrogeological extremes for the next decades. In addition, such stochastic nature of LFV does not enable any deterministic
prediction for future hydrogeological extremes on mid- and long term horizons (i.e., several years or longer). For mid- and
long-term hydrogeological extremes prediction purposes, identifying large-scale ocean-atmosphere drivers leading to such
variabilities in GWL remains fundamental. Finally, future research should investigate to what extent potential changes in the
amplitude of internal climate modes driving LFV could impact hydrogeological extremes.
**Data availability**
The groundwater level data used for this analysis can be obtained from https://ades.eaufrance.fr/ (last access: 1st April 2020).
For the database of relatively undisturbed GWL time series regarding water abstraction, contact us. The Safran precipitation
data set can be obtained from https://donneespubliques.meteofrance.fr/ (last access: 1st April 2020).



**Authors contributions**

LB, and NM conceptualized the study. LB took responsibility for the methodology, software, formal analysis, investigation, original draft preparation, and visualization. LB, NM, DA, MF, and HB validated the study. LB collected the resources and curated the data. LB, NM, DA, MF, and HB reviewed and edited the paper.

**Competing interests**

The authors declare that they have no conflict of interest.

**Acknowledgments**

This work was partially supported by the GeoERA project TACTIC, funded by the European Union's Horizon 2020 research and innovation programme under grant agreement number 731166. We would also like to thank the Agence de l'Eau Seine Normandie, PIREN Seine, BRGM and Région Normandie for their financial support. Finally, we would like to thank Sandra Lanini for the calculation of effective precipitation.

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
