# Peer review of "Influence of low-frequency variability on high and low groundwater"

_Hydrology and Earth System Sciences, 2022_

## Author Comment (AC1)

**1 Response to RC1**

We thank Referee 1 for his comments. You will find in this document responses to each comment.

*The manuscript is overall quite clear and well written. Giving more details about the novelty of the (quantitative) approach compared to the existing literature (indicated as rather qualitative) would highlight the significance of this work.*

As far as we know, the approaches in existing literature are indeed rather qualitative. The most known study on that topic for groundwater levels is that of Rust et al. (2019). In this study, they just graphically positioned the major droughts on the reconstructed low-frequency components of groundwater levels. From this, they identified that the majority of UK droughts were aligned with negative (minima) phases of the ~7-yr variability. However, there is no quantitative assessment of the importance of ~7-yr variability in the emergence of these droughts.

Bonnet et al. (2020) didn't directly study the influence of multidecadal variability on high and low groundwater levels but on high and low river flows through groundwater-river exchanges in the Seine catchment. Consequently, they only described the state (positive or negative phase) of multidecadal variability in groundwater levels to explain the occurrence and severity of two major hydrological droughts in the Seine river catchment. Hence, no quantification of the influence of multidecadal variability on high and low groundwater levels was done since it was not the final purpose of their study. We go further in our study by quantifying the dependance of high and low groundwater levels on low-frequency variability. First, we investigate the proportion of groundwater extremes resulting from the low-frequency variability in each groundwater level time series; that is, extreme levels that are generated by low-frequency components. In other words, we investigate the proportion of groundwater extremes that are not considered as "extreme levels" when one or several low-frequency components are absent from the groundwater level signal. In addition, we aim at investigating the contribution of low-frequency components in the occurrence of well-known extreme historical events by estimating a percentage of contribution in the amplitude of threshold exceedance. Such quantitative approaches improve knowledges on low-frequency induced groundwater extremes, and how much each type of aquifers, in terms of GWL variability, may be affected by such extremes, for ultimately anticipate groundwater droughts and floods.

**Abstract**

*Line 10: what do you mean by 'groundwater droughts and floods'? Droughts and floods refer to surface phenomenon. Do you mean the impact of drought and flood variability on groundwater levels?*

No we don't mean the impact of drought and flood variability on groundwater levels. By "groundwater droughts", we refer to very low groundwater levels that may lead to critical situations for anthropogenic activities (domestic, agricultural, industrial uses) or ecosystems (resulting eventually in hydrological droughts) since the water table is no longer able to sustain streamflow. By "groundwater floods", we refer to very high groundwater levels that can eventually lead to flooding due to groundwater level rise (e.g., direct emergence of groundwater at ground surface, soil/underground saturation resulting in strong overland flow).

*Lines 19 to 22: missing verb or part of the sentence, reformulate.*

We reformulate the sentence as follows: "By filtering out these components (either independently or jointly), it is possible to (i) examine the proportion of high level (HL) and low level (LL) occurrences generated by these variabilities, (ii) estimate the contribution of each of these variabilities in explaining the occurrence of major historical events associated to well-recognized societal impacts."

**Introduction**

*Line 55: it would be interesting to recall briefly the definitions of the other type of draughts and highlight how they are linked to each other or at least to the GW draught.*

We propose to add the following definitions in the revised manuscript:
- Meteorological drought refers to a precipitation deficit spanning across a large area and time-period that may be also combined with increased potential evapotranspiration.
- Hydrological drought refers to below-normal surface and subsurface water (i.e., below-normal groundwater levels, water levels in lakes, declining wetland area and decreased streamflow).
- Agricultural drought refers to a deficit in soil moisture leading to a failure in the supply of moisture to vegetation, affecting crops but also natural ecosystems.
- Socioeconomic drought refers to societal impacts of meteorological, hydrological and agricultural droughts. The water resources become insufficient to support water supply.

Meteorological droughts may lead to agricultural and hydrological droughts as it does not rain enough to (i) supply enough water to the soil, (ii) sustain streamflow. If the meteorological drought spans over a few winters, then it may create a recharge deficit over several years leading to groundwater droughts. In case of groundwater droughts and too low groundwater levels, it may also result into hydrological drought as the water table is not able to sustain streamflow. All of these droughts individually or together may lead to socio-economic drought.

*Line 65-66: this sentence is not clear. Can you reformulate?*

The response time to precipitation to get a high GWL differs a lot according to aquifer characteristics and response type (i.e., reactive vs inertial systems).

**Effective precipitation and groundwater data**

*Lines 183 to 185: I do not understand what is meant by mesh or SAFRAN mesh (temporal, spatial, spatio-temporal), and to which aim it will be used (to model which process or quantity of interest?). What is SAFRAN?*

SAFRAN is a French reanalysis providing gridded meteorological data (precipitation and snow, temperature, Penman-Monteith potential evapotranspiration) with time series spanning the 1958-2019 range at a daily timestep. For full documentation see Vidal et al. (2010). Based on such reanalysis data, we computed effective precipitation for each grid mesh with the water budget method proposed by Edijatno and Michel (1989). In a part of this study, we aimed at determining the lag between low-frequency components in effective precipitation and groundwater levels for some boreholes. For this work, we selected the mesh of SAFRAN reanalysis (spatial) with the effective precipitation time series (temporal) with the biggest correlation with GWL.

**Section 3.1**

*An illustration of the variability or energy level of the MODWT approximation might be insightful, e.g on fig 2a.*
*Would a graphical or table presentation of the MODWT and CWT results help in understanding how the variability is quantified?*

We propose to add a figure (like the figure below) in the manuscript to introduce the MODWT analysis with all components (details+last smooth) plotted and associated energy percentage to each component indicated.
As we didn't restrict the decomposition level to interior coefficients of MODWT, in order to be able to distinguish d6 (~7-yr) and d7 (~17-yr) from the last approximation (with a s20 wavelet filter), the energy level of the last approximation is 0%. All the energy is contained into wavelet details, which is shown in the figure below.

[Figure]

*Figure X: Example of multiresolution analysis by MODWT decomposition for the borehole of Tincques. In gray is presented the groundwater level time series in Tincques; in blue the wavelet details (d1 to d9) and the last smooth or approximation (s9). The Fourier period (in black) and energy percentage (in red) associated to each detail can be calculated.*

We also propose to add a CWT spectrum in the methodology section so that readers have a graphical overview of the explanations given in Section 3.1.

**Section 3.2**

*Are the 0.2 and 0.8 percentiles unique/determined for the whole dataset or by aquifer group or borehole, and over which time period?*

Percentiles 0.2 and 0.8 were determined for all boreholes of the whole dataset over the 1976-2019 period.

**Multi-timescale variability…**

*Fig. 3: it might be good to recall that the energy level is a result from the MODWT.*

It is planned to add it to the legend of the revised manuscript as follows: "Multi-timescale variability of groundwater levels in Paris Basin (78 boreholes). Pie charts describe the energy percentage of each timescale of variability reflecting their importance in total groundwater level variations. This energy percentage is derived from the MODWT analysis."

*Fig.4 caption: explain what the contour lines mean*

The contour lines express the statistically significant regions of the spectrum at a confidence level of 95% (Monte-Carlo test).

We also forgot to mention that the white transparent line corresponds to the cone of influence where the variance is underestimated due to edge effects.

*For fig 5, 6, 10 & 11 it might facilitate the reading of the results if there was an identification of the hyrdogeol entity types (iD, iMD, cAM, cA) instead of displaying the coloured hydrogeological entities.*

We think that it might facilitate the reading of the results too. We propose to keep 'aquifer types' as background for Fig. 1, Fig. 3; and to display a background corresponding to hydrogeological entity types (iD, iMD, cAM, cA) on figures that you cited.

**Discussion**

*Could you discuss the possibility of assessing the current LFV state and its short term evolution, given a type of aquifer, as it might be helpful for short term predictions of GWE?*

This is a very interesting suggestion, indeed.
For type iD aquifers, with a downward (resp. upward) phase of DV in GWL, the probability of heading for a groundwater drought (resp. flood) in the short term is higher.
For type iMD aquifers, with concomitant downward (resp. upward) phase of MAV and DV in GWL, the probability of heading for a groundwater drought (resp. flood) in the short term is higher.
For type cAM and cA aquifers, the same conclusions can be reached than iMD aquifers, however the probability of heading for a drought or flood on the short term is also highly dependent on the AV state and the last winter recharge.
However, assessing current LFV state requires adopting the right methodology, particularly if MODWT analysis is used. The resulting low-frequency components are particularly submitted to edge effects. Last values of low-frequency components are then susceptible to be biased. Consequently, it requires sufficiently long time series to extract components that are not subject to edge effects (*i.e.,* internal coefficients) and that enable the ~7-yr and ~17-yr components to be distinguished of the last approximation.

**References**

Bonnet, R., Boé, J., and Habets, F.: Influence of multidecadal variability on high and low flows: the case of the Seine basin, Hydrol. Earth Syst. Sci., 24, 1611–1631, https://doi.org/10.5194/hess-24-1611-2020, 2020.

Edijatno, and Michel, C.: Un modèle pluie-débit journalier à trois paramètres, La Houille Blanche, 2, 113–122, https://doi.org/10.1051/lhb/1989007, 1989.

Rust, W., Holman, I., Bloomfield, J., Cuthbert, M., and Corstanje, R.: Understanding the potential of climate teleconnections to project future groundwater drought, Hydrol. Earth Syst. Sci., 23, 3233-3245, https://doi.org/10.5194/hess-23-3233-2019, 2019.

Vidal, J.-P., Martin, E., Franchistéguy, L., Baillon, M., and Soubeyroux, J.-M.: A 50-year high-resolution atmospheric reanalysis over France with the Safran system, Int. J. Climatol., 30, 1627–1644, https://doi.org/10.1002/joc.2003, 2010.

---

## Author Comment (AC2)

**1 Response to RC2**

We thank Referee 2 for his comments. You will find in this document responses to each comment.

*This work investigates the effect of low-frequency variability on low and high groundwater levels extremes measured in the Paris basin. To my understandin, all methods have been applied correctly and lead to resuts which allow a thorough analysis of the "low-frequency variability induced extremes".*

*I have some minor questions and remarks.*

*L11 - At first reading it is not very clear what is meant by "sensitive". Maybe use "low GWLs are stemming/resulting from such low-frequency..".*

Indeed, the term "sensitive" is not the best one as high and low GWL may be induced by the low-frequency variability. So, we agree to change the sentence by : "It appears crucial to evaluate whether (and how much) the very high or very low GWLs are resulting from such low-frequency variability (LFV), which was the main objective of the study presented here."

*L16 - consisted in*

Yes, it will be corrected in the revised manuscript.

*L41 - I would say "in the context of global change"*

It will be corrected in the revised manuscript.

*L42 - for our societies.*

It will be corrected in the revised manuscript.

*L50 - Hydrological droughts*

It will be corrected in the revised manuscript.

*L56 - " et rates, that cause low soil moisture content.. " I am not too familiar with this, but here I do not see a difference of gw droughts compared to hydrological droughts.*

Both phenomena (hydrological droughts and groundwater droughts) are strongly linked and might even be considered the same, but groundwater droughts should actually be considered as the main driver of hydrological droughts which encompasse a wider variety of processes (see for instance Van Loon, 2015). We can add after the sentence "et rates, that cause low soil moisture content.." that groundwater droughts are characterised by decreased and below-normal GWL becoming critical to sustain human activities (agricultural, industrial or domestic uses) but also streamflow that may lead to issues in surface ecosystems.

*L65 - according to the type of aquifer and GWL variation.*

It will be corrected in the revised manuscript.

*L69 - I would say that a water level higher than the soil surface is no groundwater anymore.*

Literally yes, but such water comes from the water table and the flood is induced by the rising water table.

*L86 - "However, although these indices are useful tools to describe droughts, their principal limit arises from the standardisation allowing for spatial comparison but therefore hindering to keep the variance notion in time series" You are stating important facts here, but especially the second part is difficult to follow. I suggest to split the sentence in 2 and explain better what you mean.*

We propose to change it by: "However, although these indices are useful tools to describe droughts, their principal limit arises from the standardisation. This standardisation is useful for spatial comparison, but variance information gets lost. We equate aquifers that exhibit a weak amplitude of variations (i.e., <2m of maximum water table fluctuation) and high amplitude of variations (i.e., 10m of maximum water table fluctuation). This is particularly limiting to understand the emergence of high and low GWL whose amplitude seems highly dependent on the maximum water level fluctuation."

*L89 - dependent on*

It will be corrected in the revised manuscript.

*L107 - .. highlighted, using a composite analysis with Sea Level Pressure (SLP), that the.. (comma insertions)*

It will be corrected in the revised manuscript.

*L119 - I don't understand. Please rephrase.*

Would the sentence rephrase as following be better? "Recently, Baulon et al. (2020) also identified a significant ~17-yr variability in GWL of chalk and limestones aquifers in northern France."

*L122 - except the 1975 drought.*

It will be corrected in the revised manuscript.

*L126 - delete "to", replace "in supporting" with "on"*

It will be corrected in the revised manuscript.

*L132 - high- and low-freuquency*

It will be corrected in the revised manuscript.

*L134 - Why exaclty where these 7 and 17 year variabilies chosen? I think this needs explaining.*
*.. consists in evaluating*

Because these ~7-yr and ~17-yr variabilities have been shown to be the most important (and statistically significant) low-frequencies in hydroclimatic variables and groundwater levels in northern France and neighbouring countries (Slimani et al., 2009; Massei et al., 2010; Rust et al., 2019; Baulon et al., 2022 (see Fig. 7 with global wavelet spectra)).

*140 - Rephrase please*

"Second, we propose determining on four well-known historical events the contribution of multi-annual and decadal variabilities in the amplitude of threshold exceedance (ATE) and identify what parameters may control this contribution. In other words, we estimate the percentage of contribution of each low-frequency component in the emergence of the historical event."

*L145 - Boreholes were selected from a BRGM database and were required to be undisturbed from human acitivites. We selected the boreholes by following the three steps below.*

It will be corrected in the revised manuscript.

*L149 - the removal of pre-selected..*

Not the "removal" but "cross-referencing". We checked in the BRGM databases if there were known anthropogenic influences on the pre-selected boreholes.

*L170 - Do you mean sub-monthly?*

No, we don't mean sub-monthly. We are not interested by such high-frequency variations, rather mostly in annual variations (and beyond) of GWL that can be particularly significant in aquifers with reactive GWL variations.

*L190 - were -> was*

It will be corrected in the revised manuscript.

*L214 - for that purpose*

It will be corrected in the revised manuscript.

*L245 - But isn't this also a consequence of the initial amplitude of exceedance?*

Yes it is. But here we consider that a low-frequency component influences an extreme level if the component generates the extreme level or if the level is no longer considered as an extreme if the component is subtracted from the GWL signal. In other words:
- if the peak still exceed the threshold when the LFV is subtracted, the peak remains an extreme level, and then we considered that the subtracted component has no influence on extreme emergence.
- if the peak no longer exceed the threshold when the LFV is subtracted, the peak is then no longer considered as an extreme level, and we considered that the substracted component significantly influences the extreme emergence, since it generates the extreme level.

For better clarity, we propose to introduce the above explanation into this section (3.2.) of the manuscript.

*Fig3 - I prefer the term variance, but this is a matter of preference.*

We used this term "energy" as it is the term used by Constantine and Percival (2016).

*Fig9 - Why not splitting the period exactly in half, doing the analsis for both periods and the complete period 1903-2019?*

We presented results as such because the aim was to compare results obtained over the 1976-2019 period with those obtained on the entire time series (i.e., 1903-2019).

*P23 - 28 : I wonder how the results are sensitve to the choice of 7/17 and 7&17 year variablilty. I think this has to be discussed or at least commented on.*

The ~7-yr and ~17-yr variabilities have been specifically chosen because previous studies showed that these components were the dominant low-frequency variabilities (and statistically significant) in hydroclimatic variables, including in groundwater levels, in northern France and neighbouring countries (e.g., Massei et al., 2010; Rust et al., 2019). This is why we are interested in these frequencies in this study and not others.

*L582 – typo*

"strengthen": it will be corrected in the revised manuscript.

*L584 - linked to -> resulting from*

It will be corrected in the revised manuscript.

*L589 - dating back to*

It will be corrected in the revised manuscript.

*L595 - Which aperiodic behaviour? I don't understand this sentence*

Here, we talk about the aperiodic behaviour of low-frequency variabilities. Indeed, they are not periodic (phase and amplitude are not constant over time) even though they look more or less periodic. We propose to correct the sentence as such: "Knowing the dependence of GWE to LFVs, this aperiodic behaviour of LFVs can heavily influence the HL and LL severity in aquifers displaying inertial or combined GWL variation types."

*L601 - largely -> generally*

It will be corrected in the revised manuscript.

**References**

Baulon, L., Allier, D., Massei, N., Bessiere, H., Fournier, M., and Bault, V.: Influence de la variabilité basse-fréquence des niveaux piézométriques sur l'occurrence et l'amplitude des extrêmes, Géologues, 53–60, 2020.

Baulon, L., Allier, D., Massei, N., Bessiere, H., Fournier, M., Bault, V.: Influence of low-frequency variability on groundwater level trends, J. Hydrol., 606, 127436, https://doi.org/10.1016/j.jhydrol.2022.127436.

Constantine, W., and Percival, D.: wmtsa: Wavelet Methods for Time Series Analysis. R package version 2.0-1. <https://CRAN.R-project.org/package=wmtsa>, 2016.

Massei, N., Laignel, B., Deloffre, J., Mesquita, J., Motelay, A., Lafite, R., and Durand, A.: Long-term hydrological changes of the Seine River flow (France) and their relation to the North Atlantic Oscillation over the period 1950–2008, Int. J. Climatol., 30, 2146–2154, https://doi.org/10.1002/joc.2022, 2010.

Rust, W., Holman, I., Bloomfield, J., Cuthbert, M., and Corstanje, R.: Understanding the potential of climate teleconnections to project future groundwater drought, Hydrol. Earth Syst. Sci., 23, 3233-3245, https://doi.org/10.5194/hess-23-3233-2019, 2019.

Slimani, S., Massei, N., Mesquita, J., Valdés, D., Fournier, M., Laignel, B., and Dupont, J.-P.: Combined climatic and geological forcings on the spatio-temporal variability of piezometric levels in the chalk aquifer of Upper Normandy (France) at pluridecennal scale, Hydrogeol. J., 17, 1823, https://doi.org/10.1007/s10040-009-0488-1, 2009.

Van Loon, A. F.: Hydrological drought explained, WIREs Water, 2, 359–392, https://doi.org/10.1002/wat2.1085, 2015.